# Holistic Approach to the Restoration of a Vandalized Monument: The Cross of the Inquisition, Seville City Hall, Spain

**Valme Jurado** [1], **Juan Carlos Cañaveras** [2], **Antonio Gomez-Bolea** [3], **Jose Luis Gonzalez-Pimentel** [1], **Sergio Sanchez-Moral** [4], **Carlos Costa** [5] and **Cesareo Saiz-Jimenez** [1,*]

1 Instituto de Recursos Naturales y Agrobiologia, IRNAS-CSIC, 41012 Sevilla, Spain; vjurado@irnase.csic.es (V.J.); pimentel@irnas.csic.es (J.L.G.-P.)
2 Departamento de Ciencias de la Tierra, Universidad de Alicante, 03080 Alicante, Spain; jc.canaveras@ua.es
3 Departament de Biologia Evolutiva, Ecologia i Ciències Ambientals, Facultat de Biologia, Institut de Recerca de la Biodiversitat (IRBio), Universitat de Barcelona, 08028 Barcelona, Spain; agomez@ub.edu
4 Museo Nacional de Ciencias Naturales, MNCN-CSIC, 28006 Madrid, Spain; ssmilk@mncn.csic.es
5 Atelier Samthiago, 4900-374 Viana do Castelo, Portugal; ccosta@samthiago.com
* Correspondence: saiz@irnase.csic.es

**Abstract:** The Cross of the Inquisition, sculpted in 1903 and raised on a column with a fluted shaft and ornamented with vegetable garlands, is located in a corner of the Plateresque façade of the Seville City Hall. The Cross was vandalized in September 2019 and the restoration concluded in September 2021. A geological and microbiological study was carried out in a few small fragments. The data are consistent with the exposure of the Cross of the Inquisition to an urban environment for more than 100 years. During that time, a lichen community colonized the Cross and the nearby City Hall façades. The lichens, bryophytes and fungi colonizing the limestone surface composed an urban community, regenerated from the remains of the original communities, after superficial cleaning of the limestone between 2008 and 2010. This biological activity was detrimental to the integrity of the limestone, as showed by the pitting and channels, which evidence the lytic activity of organisms on the stone surface. Stone consolidation was achieved with Estel 1000. Preventol RI80, a biocide able to penetrate the porous limestone and active against bacteria, fungi, lichens, and bryophytes, was applied in the restoration.

**Keywords:** green algae; lichens; *Trebouxia aggregata*; black fungi; bryophytes; limestone; mineralogy; restoration; Seville City Hall

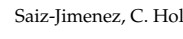



## 1. Introduction

The construction of the Seville City Hall building started in 1526 and the last works ended in 1928. The main façade, facing to Plaza Nueva was completed in 1867. The Plateresque façade from the 16th century, facing to Plaza de San Francisco, was continued in the 19th and reliefs were carved between the end of the 19th and the 20th centuries. Still today, part of this façade is unfinished. In a corner of the Plateresque façade is located the Cross of the Inquisition, sculpted in 1903, raised on a column with a fluted shaft and ornamented with vegetable garlands (Figure 1A). Previously, another Cross, dating from the end of the 18th century, devoid of ornamentation, was placed in the same position as a commemoration of the last case of death at the stake (year 1781), although a few historians believe that the primitive Cross was erected during a plague epidemic in the 16th century. The Cross and the City Hall façade were restored between 2008 and 2010, and were vandalized in the early morning of 10 September 2019 (Figure 1B). The Cross fragments were stored for two years until a restoration was decided, which should include the repair of damages caused by the act of vandalism, but also previous pathologies caused

by environmental pollution and anthropic actions which resulted in cracks and losses of materials. The restoration concluded last September 2021 (Figure 1C), and included a geological and microbiological study that, unfortunately, was only possible on a few fragments. The limestone fragments showed dark stains and crusts, likely originated by lichen colonization. Lichen communities were found on the nearby City Hall limestone. Here, we present the data obtained in the study, which were to some extent limited by the scarce availability of samples, as all fragments should be used in the restoration, as well as the time that the fragments were kept in storage which prevented the study of fresh biological samples.

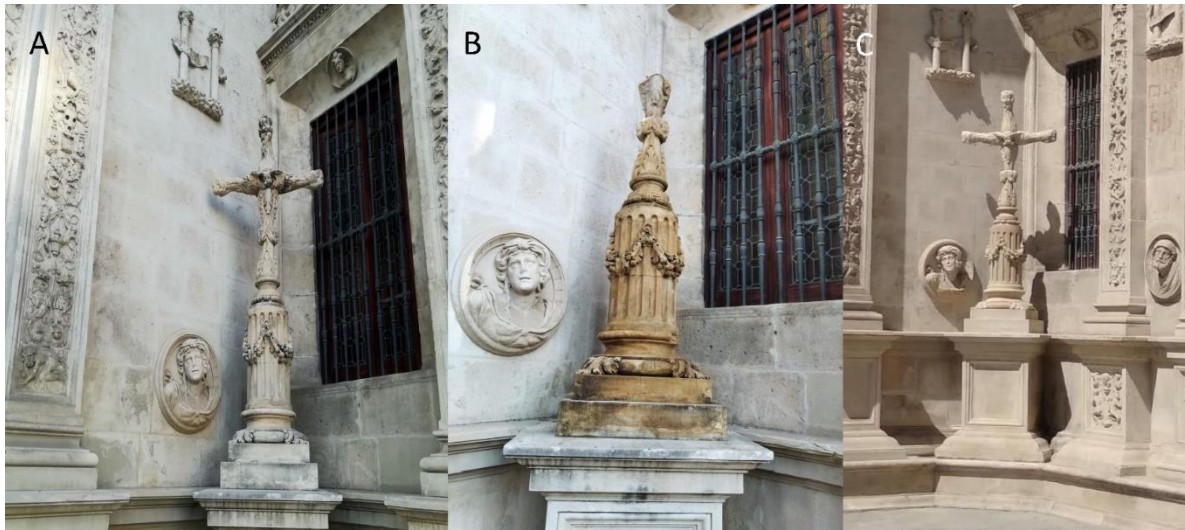

**Figure 1.** The Cross of the Inquisition before (**A**), after the vandalism (**B**), and restored (**C**).

## 2. Materials and Methods

Six small samples, three for geological and three for microbiological studies, were collected under the supervision of the restorers (Figure 2).

Mineralogy and textural properties of samples were studied on (30 μm-thick) polished thin sections on a transmitting light microscope. A staining procedure with combined alizarine red S and potassium ferricyanide solution was applied to distinguish ferroan and non-ferroan calcite and dolomite phases [1]. Two thin sections were made from the sample, one from the external surface of the sample and the other being transverse to that surface. Photomicrographs were performed by using a petrographic microscope: Zeiss Assioskop, a digital camera: USB UI-1490SE and an image capture software: uEye Cockpit (IDS). The sample was also studied under a scanning electron microscope (SEM) using BSE mode on a Hitachi S3000N SEM coupled with an X-ray detector, Bruker XFlash 3001, for microanalysis (EDS) and mapping. The samples' observation was carried out under variable pressure mode without coating with any conductive material. EDS analyses were performed at a 40 Pa chamber atmosphere, 20 kV accelerating voltage and 10–12 mm working distance.

The mineralogical composition was analyzed by powder X-ray diffraction in a Bruker D8 Discover A25 microdiffractometer. Microdiffraction X-ray equipment makes it possible to make precise measurements in very small areas without the need to prepare the sample. The analysis area was selected using an optical microscope and an X-ray micro-source with a copper anode, using its Kα spectral line. EVA and TOPAS computer programs were used for data processing and qualitative and quantitative identification, with the Rietveld method, of the crystalline phases present in the sample.

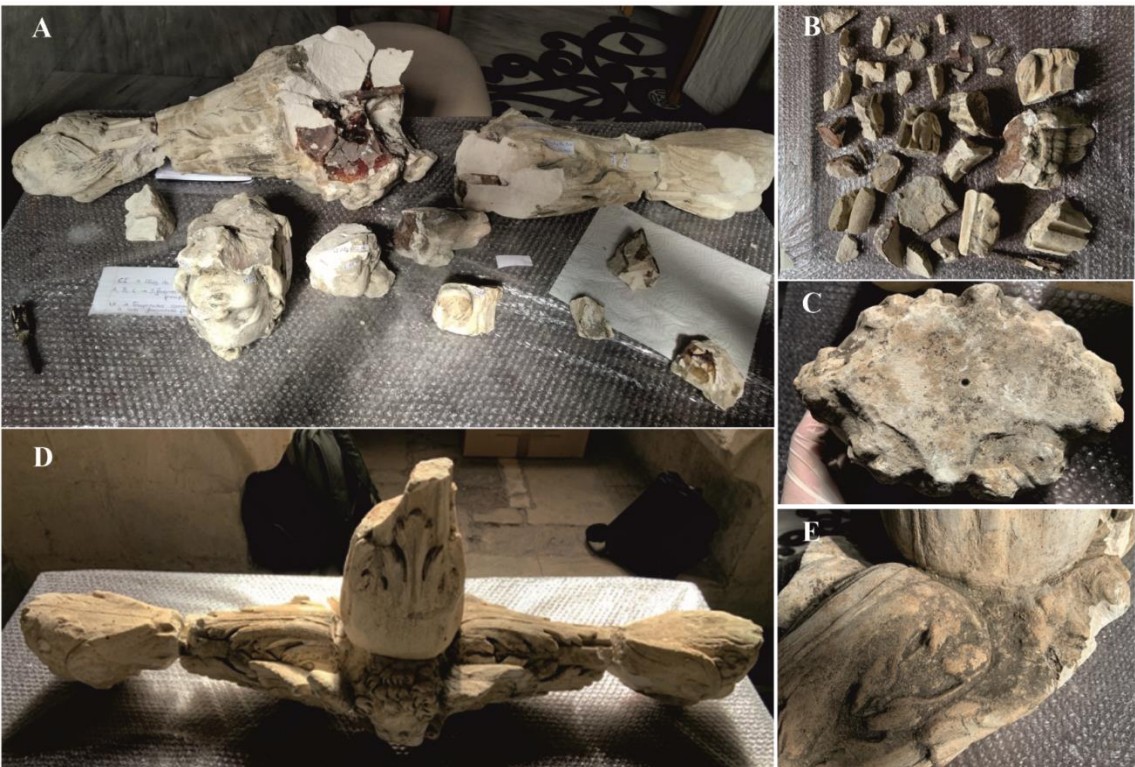

**Figure 2.** Vandalization of the Cross of the Inquisition. (**A,B**) Fragments of different sizes. (**C,E**) Fragments with black to brown crusts. (**D**) Reconstruction of some fragments.

For the identification of the organisms two approaches were adopted in this study. First, due to the fact that the vandalized fragments of the Cross were stored for two years, and no fresh samples from organisms could be obtained, we focused our attention on the identifications of DNA on the fragments available, in order to determine past biological colonizations. In addition, we sampled the lichen community colonizing the City Hall limestones in an area around 5 m from the Cross. It was expected that the community or at least the most abundant lichens in both the building façade and Cross would be the same. The protocols used were as follows:

Nucleic acids from each sample were extracted with commercial extraction kits, following the manufacturer's instructions. FastPrep matrix-E lysis tubes (Qbiogene, Carlsbad, CA, USA) with glass beads were used and physical disruption was performed using a shaker (Fast Prep-24, Solon, OH, USA) at a speed of stirring at 5.5 m/s for 2 × 30 s. The extracted nucleic acids (50 µL for each sample) were stored at −80 °C until shipment for sequencing. The concentration of the DNA extracted from the samples was measured by fluorometric quantification using a Qubit 2.0 fluorimeter (Invitrogen, Carlsbad, CA, USA). Eukaryotes were identified by sequencing the 18S ribosomal RNA gene. The genes were amplified by PCR using the primer pairs EUKA (5′-AAC CTG GTT GAT CCT GCC AGT-3′) and EUKB (5′-TGA TCC TTC TGC AGG TTC ACC TAC-3′) for eukaryotes.

Amplifications were carried out in an iCycler thermal cycler from BioRad (Hercules, CA, USA). Reactions were set up in 0.2 mL tubes (Greiner BioOne, Monroe, LA, USA). Each reaction contained: 5 µL of 10× BioTaq PCR buffer (Bioline, Randolph, MA, USA), 1.5 µL of MgCl2 (50 mM stock solution), 1 µL of a mixture of the four nucleotides (dNTP) (2 mM) (Invitrogen, Carlsbad, CA, USA), 0.5 µL of each of the primers with a concentration of 50 µM, 0.25 µL of the equivalent BioTaq DNA polymerase (Bioline, Randolph, MA, USA) at 2.5 units and template DNA. The final volume of the reaction was made up to 50 µL with ultrapure water. For each reaction, about 10–20 ng of DNA extract was added. The PCR protocol used was as follows: 5 min of initial denaturation at 94 °C; 30 cycles consisting of 2 min at 94 °C, 15 s at 55 °C, and 2 min at 72 °C; followed by a 10-min final

extension at 72 °C. The results of PCR reactions were verified by horizontal electrophoresis in agarose gels (1% *w/v*) and the amplified products were purified using the JETquick Spin kit (Genomed, Bad Oeynhausen, Germany), and stored at −20 °C.

In order to identify the main organisms that were present in the limestone samples, libraries were constructed from the purified PCR products of the 18S rRNA genes. For this purpose, purified products were cloned using the commercial TOPO TA Cloning Kit for Sequencing kit (Invitrogen, Carlsbad, CA, USA) following manufacturer's instructions. The vector used was the plasmid pCR4-TOPO (Invitrogen, Carlsbad, CA, USA). The colonies obtained were picked at random using sterile toothpicks and grown in LB liquid medium with ampicillin. In order to verify that the collected clones had the insert (purified PCR product), a PCR was performed using the primers promoter T7 (5′-TAA TAC GAC TCA CTA TAG GG-3′) and M13 Reverse (5′-CAG GAA ACA GCT ATG AC-3′), which were specific to the vector sequence. This amplified the complete sequence of the insert and part of the vector at the ends. The clones were processed at the company Secugen (CIB-CSIC, Madrid, Spain), with an ABI 3700 capillary sequencer (Applied Biosystems, Foster City, CA, USA).

The sequences obtained were edited with the BioEdit 7.0.5.3 program. Sequences that were too short (<300 nucleotides) or whose chromatogram showed poor quality were directly removed. All non-redundant database sequences were compared with sequences deposited at the National Center for Biotechnology (NCBI) using the BLASTn algorithm [2]. We only considered sequences above 94.5%, which is the minimum threshold sequence identity for confirming a genus affiliation [3]. The nucleotide sequences generated in this study were deposited into the NCBI GenBank database under accession numbers ON479828–ON479853.

## 3. Results and Discussion

### 3.1. Mineralogical, Chemical and Petrophysical Characteristics of the Stone

The stone, being massive and granular in appearance, is not very compact, with a high porosity and a medium–low density, and exhibits a thin altered layer or surface crust. It is white to light beige (N9 to 10YR 8/2, Munsell rock color chart) with dark mottling in the altered surface area.

XRD analysis showed a predominant calcite mineral composition (90–95%), as well as dolomite (5–10%). Analysis of the surface of the sample by μXRD also revealed the presence of calcite as the main constituent, and other accessory minerals, such as thenardite ($NaSO_4$) and hematite-type iron oxides ($Fe_2O_3$).

The stone is a medium- to coarse-grained oolitic/peloidal limestone. According to the most used limestone classifications, it would be classified as oopelsparite [4] and oolitic/peloidal grainstone [5]. It is composed of medium- to coarse-sized (0.15–1 mm), rounded to sub-rounded grains that show poor sorting (Figure 3), developing a grain-supported fabric with point, long and concave–convex contacts between the grains (Figures 3 and 4A).

Highly micritized ooids and peloids (85–95%), ranging from 150 to 750 μm in size, (Figures 3 and 4) are the predominant framework constituents. Echinoderms, molluscs and foraminifera fragments of highly variable size were also observed, reaching 2 mm thick in some cases (Figure 3A–D). The bioclasts showed a good state of conservation in the case of echinoderm plates and are quite micritized in the rest of bioclasts (Figures 3 and 4). Non-carbonate grains are very scarce, mainly consisting of small grains of silicate (Figure 3F) or iron oxide minerals. A very scarce interstitial matrix (micrite) between framework grains was observed. The main cement types recognized were micro- and mesocrystalline calcite cements, showing both continuous circumgranular and occluding (syntaxial, mosaic, poikilotopic) geometries in intergranular positions, corresponding to different cementation phases (Figures 3C–F and 4A). Micro-mesocrystalline cements partially filling intraparticle pores (Figure 3C,D), and microcrystalline cements filling fractures or veins (10–25 μm

thick) (Figure 4B) were also observed. Locally, some microcrystalline siliceous cement was recognized in an interparticle position (Figure 3F).

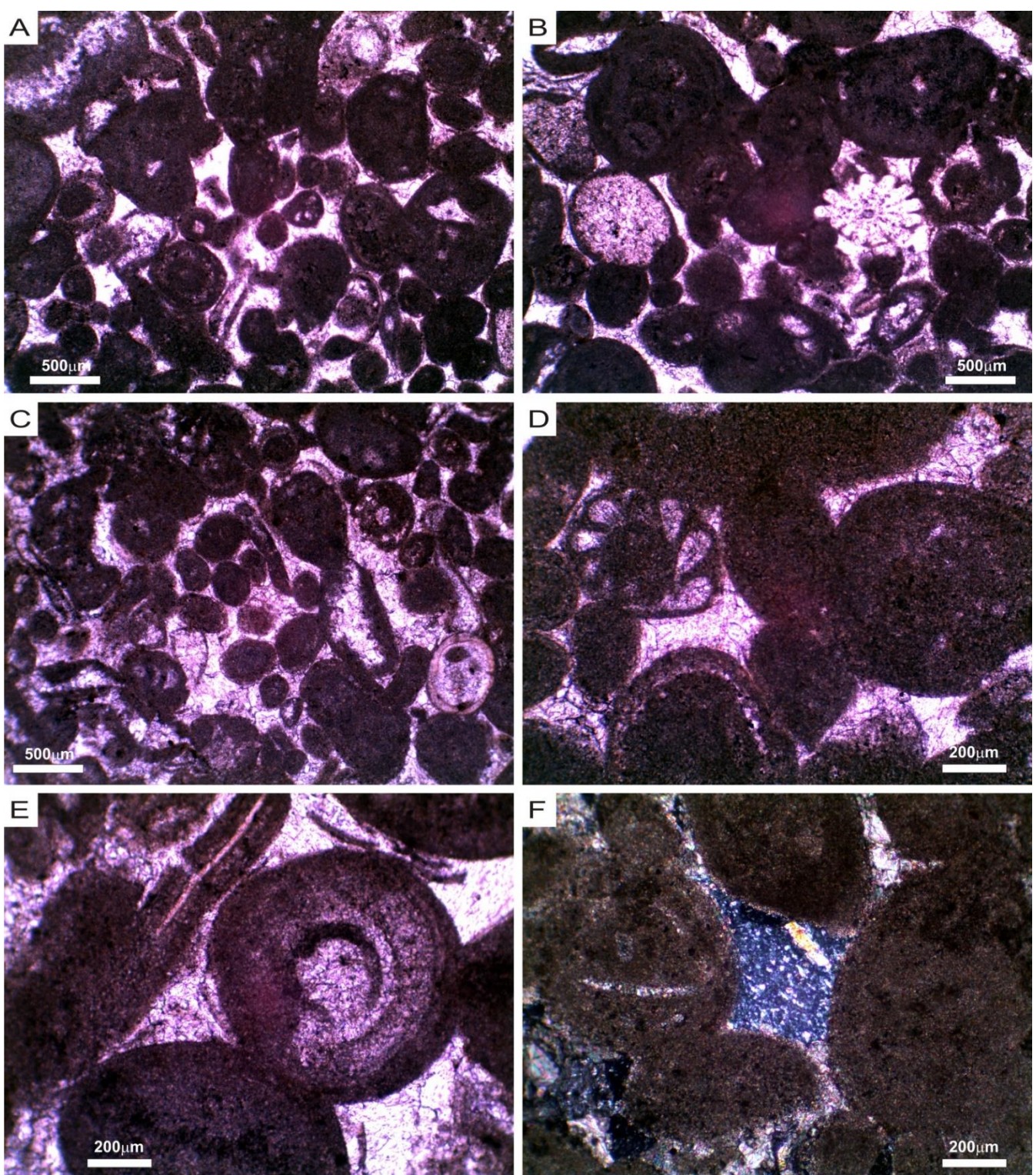

**Figure 3.** Optical photomicrographs. (**A**–**C**) General view of the sample showing a grain-supported fabric with framework composed of ooids, peloids and bioclast fragments, all of them bound by intercrystalline calcite cements. (**D**,**E**) Interparticle and intraparticle calcite cements between grains showing point, long and concave–convex contacts. (**F**) Detail of altered silicate grain and microcrystalline siliceous cement. (**A**–**E**) Plane polarized light. (**F**) Cross polarized light.

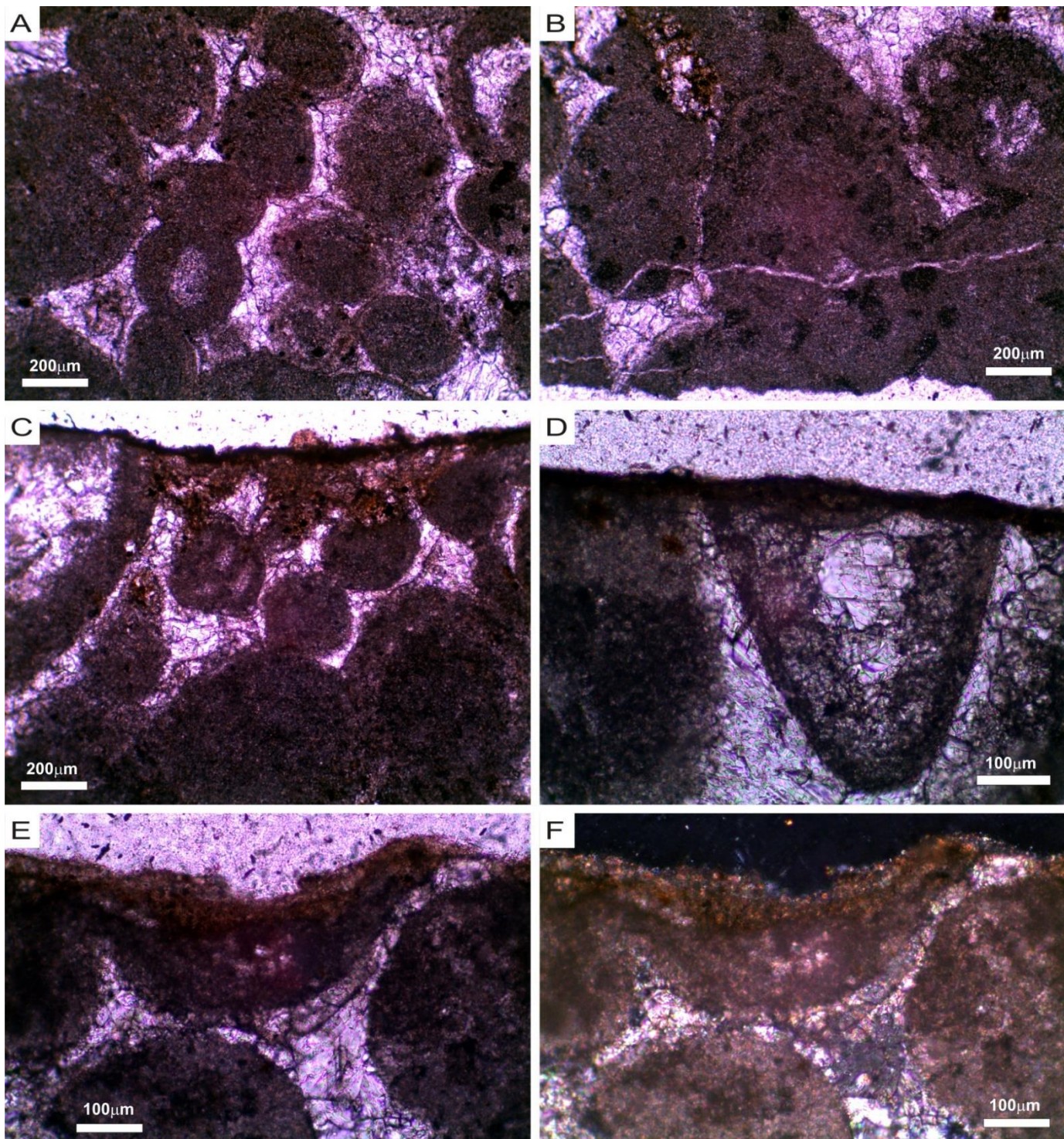

**Figure 4.** Optical photomicrographs. (**A**) Interparticle and intraparticle calcite cements between grains with point, long and concave–convex contacts. (**B**) Detail of a fracture filled by microcrystalline calcitic cement. (**C**) General view of the surface alteration crust. (**D–F**) Detail of the surface alteration crusts (see text). (**A–E**) Plane polarized light. (**F**) Cross polarized light.

The porosity of the sample is relatively high (5–15%) and fabric selective, mainly with interparticle and intercrystalline pore types. Framework grains (ooids, bioclasts) showed severe micritization processes (Figures 3 and 4). Most of the ooids appeared as rounded micritic grains that locally presented relics of primary microstructures (laminae,

banded, internal chambers). The sample showed evidence of mechanical and/or chemical compaction (long and concave–convex contacts) (Figures 3D,E and 4A), as well as fracturing (Figure 4B).

The surface alteration crust showed a variable thickness (20–100 μm) (Figure 4C–F), a compact texture, a predominantly very small crystal/grain size (<25 μm), and a greater abundance of oxides.

### 3.2. Scanning Electron Microscopy and Microanalysis (SEM-EDS)

The surficial limestone appeared to be covered by a thin, darker layer with gray to black mottling, commonly discontinuous (Figure 5). The components (grains, crystals) that compose the surface layer include iron oxides, salts (sodium and calcium sulfates, sodium chloride) and carbonaceous particles, as well as organic components (biofilm, microorganisms). Salt crystals were also detected in more internal areas, in the porous system (Figure 5D), which contributes to the granular disintegration of the stone [6–9].

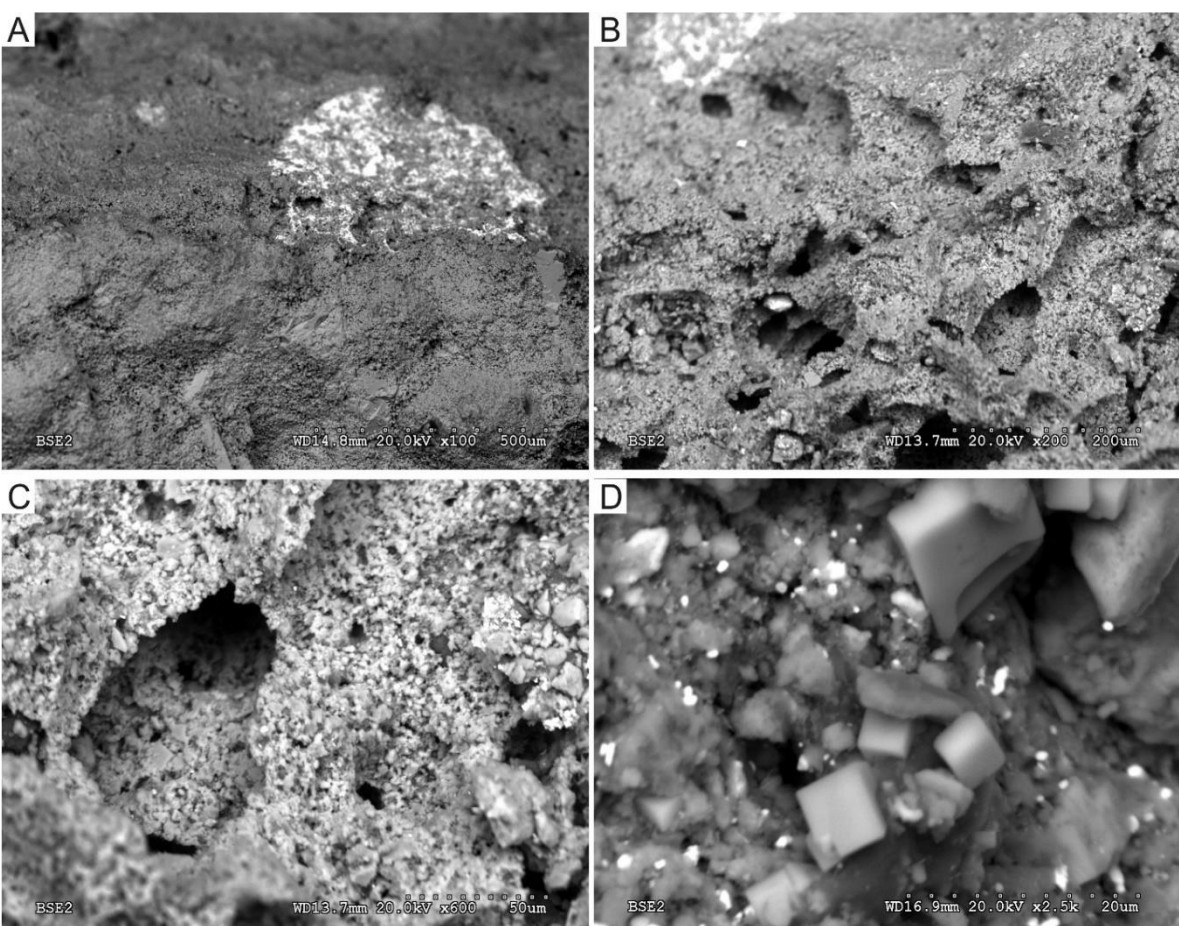

**Figure 5.** SEM micrographs showing the limestone surface of the Cross of the Inquisition. (**A**) General view of limestone surface showing zones with preferential Fe-staining (mottling). (**B**,**C**) Detail of the high porosity (interparticle and intercrystalline pores) at the sample's surface. (**D**) Detail of small idiomorphic halite (NaCl) crystals grown in interparticle porosity.

SEM-EDS analysis showed that the chemical composition of superficial layer is variable and includes (atom.%): O (66.3–64.93), C (11.64–17.44), S (1.38–5.42), Cl (0.16–0.26), Si (1.59–3.13), Al (0.61–0.92), Ca (5.56–8.63), Mg (1.17–1.53), Na (0.23), K (0.22–0.30), Ba (1.15–5.40), Fe (0.42–0.67) and Zn (0.14–0.41).

These data point to the existence of silicates (clays), soot and sulfated minerals (gypsum, thenardite) in addition to the observed halite, which seem to indicate a strong impact

due to environmental contamination from vehicle exhaust pipes. The effects of this contamination were described in the Cathedral of Seville, near the City Hall [10–12]. In some stone surfaces, a deep deterioration caused by biological colonization, with abundant pitting and channels produced by organisms, was observed (Figure 6).

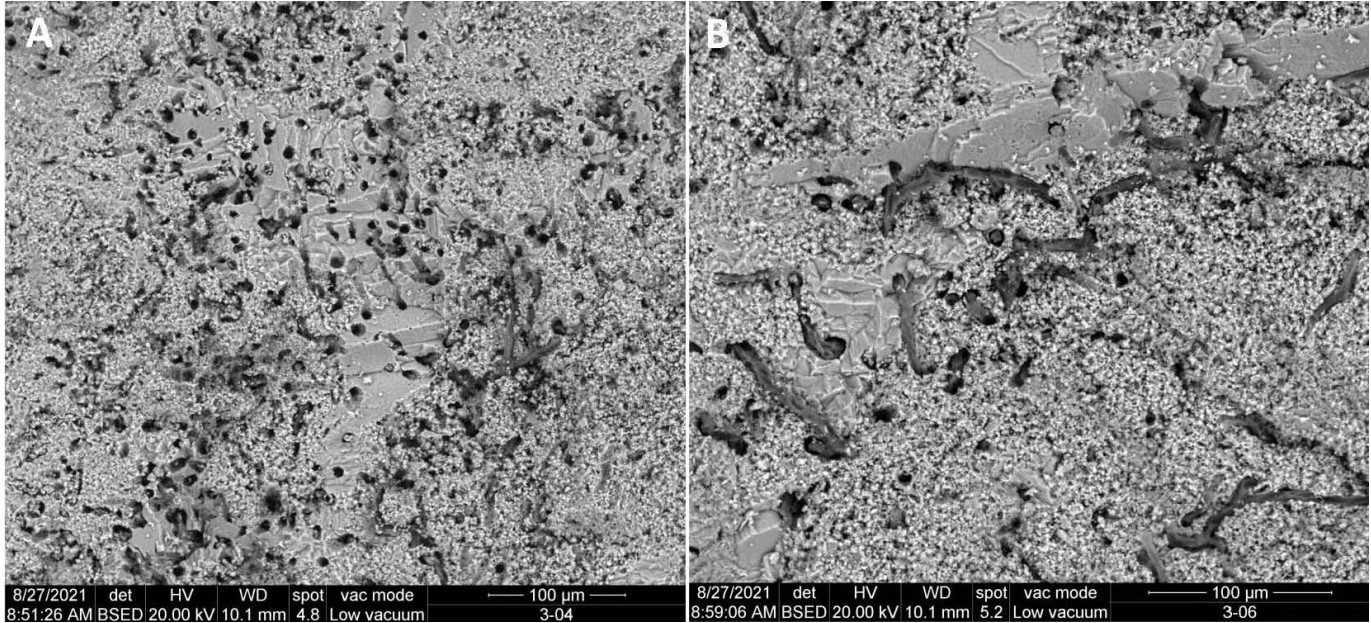

**Figure 6.** SEM micrographs showing biological weathering features on the limestone surface, Cross of the Inquisition. (**A**) Pits generally show a cylindrical morphology, a diameter between 2 and 10 μm and are not interconnected, although transitions patterns to interconnected pits can also be observed. (**B**) Channels produced by the action of filamentous microbes on the limestone surface.

Indeed, Figure 6A shows the abundance of pitting in the limestone. Pitting is generally produced by cyanobacteria, algae and fungi, and has been widely described as a product of the dissolution of limestone caused by the biological activity of colonizing organisms [13–15]. The existence of pitting, without biological structures inside, as observed in Figure 6A, denotes that the colonization was ancient, and the organisms have disappeared.

In addition to pitting, the dissolution of the limestone is produced by hyphae up to 3 microns in diameter, possibly from fungi, which are shown in Figure 6B, confirming that the colonization, in this case, was more recent.

### 3.3. Organisms on the Cross of the Inquisition limestone

Table 1 shows a summary of the eukaryotic organisms found in the limestone. Table 1 highlights the occurrence of *Trebouxia aggregata* in the black and brown crusts, which clone sequences show over 99% identity. *Trebouxia aggregata* is a unicellular green alga found in any habitat and is the most common photobiont of lichens. The green alga *Trebouxia* is the photobiont of many lichens such as *Lecanora, Caloplaca, Xanthoria, Ramalina, Buellia, Umbilicaria*, etc. [16–19].

The occurrence of *Trebouxia aggregata* suggests that the black crust observed on the limestone was actually lichens. The storage of the Cross fragments for two years prevented the collection of fresh material for an accurate taxonomical identification of the lichen community.

**Table 1.** Description of samples from the Cross of the Inquisition, Seville City Council, and list of eukaryotic organisms.

| Sample | Species | Accession Number | Group * |
| --- | --- | --- | --- |
| Brown crust (CD1) | *Trebouxia aggregata* | ON479828 | *Chlorophyta* |
| | *Tortula truncata* | ON479827 | *Bryophyta* |
| | *Pseudostichococcus monallantoides* | ON479830 | *Chlorophyta* |
| | *Phoma* sp. | ON479831 | *Fungi* |
| | *Naganishia albida* | ON479825 | *Fungi* |
| | *Myrmecia* sp. | ON479826 | *Chlorophyta* |
| | *Alternaria alternata* | ON479829 | *Fungi* |
| Black crust (CD3) | *Scleroconidioma sphagnicola* | ON479833 | *Fungi* |
| | *Pseudostichococcus monallantoides* | ON479834 | *Chlorophyta* |
| | *Neofusicoccum parvum* | ON479835 | *Fungi* |
| | *Lichinella cribellifera* | ON479836 | *Lichen* |
| | *Aureobasidium pullulans* | ON479837 | *Fungi* |
| | *Phoma* sp. | ON479832 | *Fungi* |
| Black crust (LQ1) | *Trebouxia aggregata* | ON479838 | *Chlorophyta* |
| | *Pseudostichococcus monallantoides* | ON479839 | *Chlorophyta* |
| | *Dothidea berberidis* | ON479840 | *Fungi* |

* Lichens are taxonomically classified by their fungal partner, so all lichens belong to the *Fungi* kingdom. However, in this work we distinguish between fungi and lichens for convenience.

The green alga *Pseudostichococcus monallantoides* appeared in the three crusts (Table 1). This microalga is halotolerant and salt tolerance mechanisms are associated with its resistance to desiccation, a necessary behavior for microalgae in terrestrial environments [20]. *Myrmecia* sp., a green alga in the brown crust, is frequent in desert biological crusts [21], and is a lichen photobiont [22].

Together with the green algae, the bryophyte *Tortula truncata*, synonymy of *Pottia truncata*, was identified in the brown crust. Casas et al. [23] considered that this species is rarely distributed in Spain, although specimens collected in the provinces of Seville and Badajoz are deposited in the Herbarium of the Faculty of Sciences, University of Oviedo.

*Tortula* species are usually common on the walls of buildings in urban environments [24], where they tend to resist air pollution and periods of desiccation. Isermann [25] found *Tortula truncata*, *Tortula muralis*, *Syntrichia ruralis* and many other species of bryophytes in the campus of Bremen University. Ekwealor and Fisher [26] found *Syntrichia caninervis* and *Tortula inermis*, with a high tolerance to desiccation, under the rocks of the Mojave Desert, demonstrating their resistance in hypolithic habitats.

Table 1 comprises the fungi *Phoma* sp., *Naganishia albida*, *Alternaria alternata*, *Scleroconidioma sphagnicola*, *Neofusicoccum parvum*, *Aureobasidium pullulans*, and *Dothidea berberidis*. Most of these fungi form the group of dematiaceous fungi, which are characterized by their black color due to the synthesis of melanin and by inhabiting rocks in arid environments [27,28]. The high identity values of the clone sequences allow for the identification at the genus and species level.

The genus *Phoma* comprises saprophytic and many other lichenicolous species [29,30]. *Phoma* spp. have been isolated from the lichens *Caloplaca, Cladonia, Ramalina*, etc. [31]. *Phoma* is frequent on rock surfaces in urban environments, with *Phoma glomerata* being the most abundant species.

*Naganishia albida*, a saprophytic yeast found in the Cross, can be isolated from different environments such as air, soils, bryophytes and plants [32,33]. *Alternaria alternata*, a cosmopolitan species, is one of the most frequent fungi in the air, stones and plants [34,35].

*Scleroconidioma sphagnicola* is a necrotrophic parasite of bryophytes [36,37]. *Neofusicoccum parvum*, a common and cosmopolitan plant pathogen species, has been isolated from a wide variety of hosts [38,39].

*Aureobasidium pullulans* is a ubiquitous species of black yeast, saprophytic in plants and frequent in the air of urban environments [40], but also found on monument surfaces [41–43].

*Dothidea berberidis* is included in the order *Dothideales,* comprising melanized fungi from rocks [44]. *Aureobasidium pullulans* also belong to this order.

Table 1 also comprises *Lichinella cribellifera*, a lichen growing on dry rock surfaces and representative of the xerophilous lichen flora of Southern Spain [45,46].

Rock-inhabiting fungi usually colonize the surfaces, forming a dark patina adhered to the substratum [47–50]. Electron microscope observations demonstrated the presence of pitting, whose shape and size were compatible with the endolithic action of filamentous microorganisms, observed in the Cross limestone (Figure 6A). These fungi are characterized by enduring extreme changes in humidity and long periods of desiccation, which undoubtedly is an advantage in the colonization of limestone in the climate of Seville.

Sterflinger and Prillinger [51] studied the buildings of Vienna, finding a diversity of fungi greater than in the same types of rocks in rural environments. Among the fungi, three species of *Phoma* were isolated: *Phoma exigua* var. *foveata*, *Phoma glomerata*, *Phoma macrostoma* and *Aureobasidium pullulans*, among others. *Aureobasidium pullulans* and *Phoma* sp. were identified in the Cross, which suggests a similar ecology of these species in limestone monuments in urban environments.

To discern the origin of the photo- and mycobionts listed in Table 1 we surveyed the lichens colonizing the City Hall façade, close to the Cross location. Three main lichen species were abundant on the façade, with white, yellow and blackish brown thalli (Figure 7).

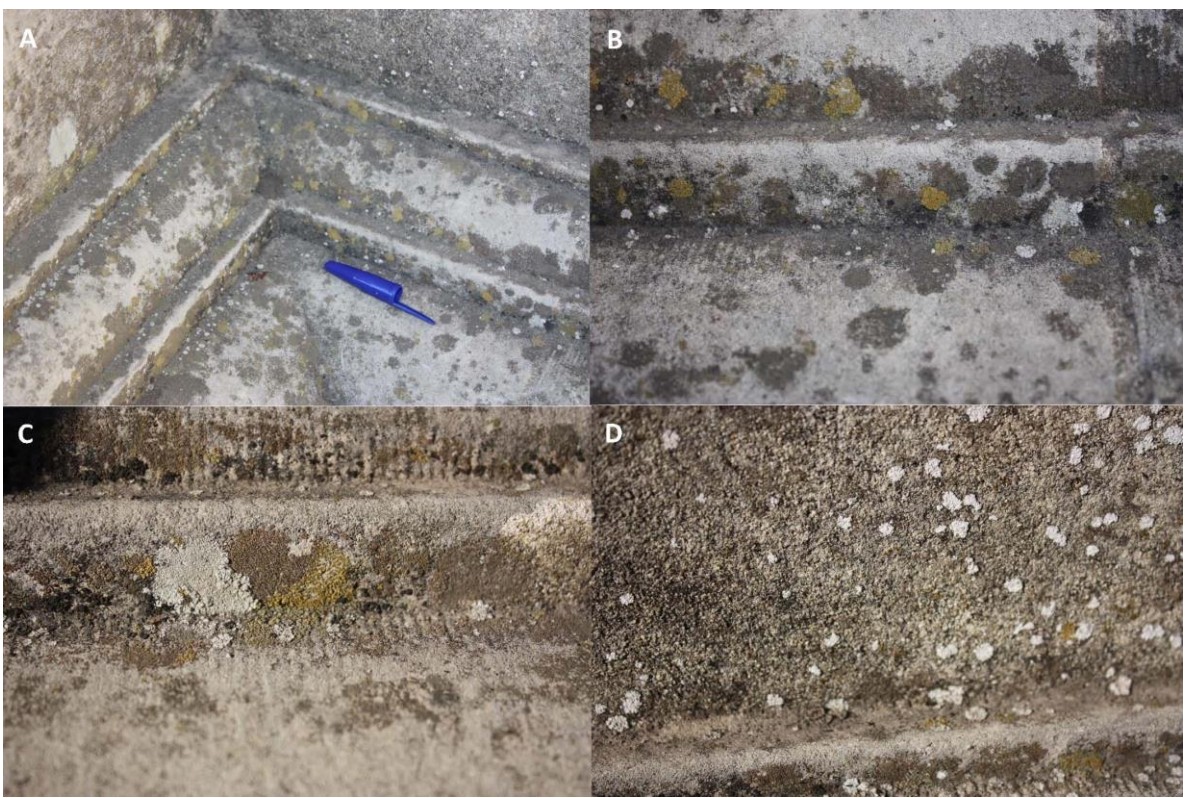

**Figure 7.** Lichen communities on the City Hall façade, close to the Cross of the Inquisition. (**A**) Community dominated by *Pyrenodesmia variabilis*. (**B**) Same community with *Caloplaca citrina* group species. (**C**) Community of *Pyrenodesmia variabilis*, *Caloplaca citrina* group species and *Kuettlingeria teicholyta*. (**D**) Community with *Kuettlingeria teicholyta*.

The white lichen (Figure 7D) was identified as being *Kuettlingeria teicholyta* (Ach.) Trevis. (=*Caloplaca teicholyta* (Ach.) J. Steiner). The yellow lichen (Figure 7B,C) was included in the *Caloplaca citrina* group [52]. This species complex, which still needs a thorough revision in Spain, occurs in a wide variety of substrata, from asbestos–cement, concrete and mortar, to basic siliceous rocks or even wood, and is very tolerant to, and even

favored by eutrophication [53]. The blackish brown lichen (Figure 7A,B) was affiliated with *Pyrenodesmia variabilis* (Pers.) A. Massal. (=*Caloplaca variabilis* (Pers.) Th. Fr.). Taxonomical reviews on the family *Telochistaceae* transferred species of the genus *Caloplaca* to the new genera *Kuettlingeria* and *Pyrenodesmia* [54,55]. Muggia et al. [56] revised the black endolithic *Caloplaca* and reported that almost all the species shared *Trebouxia* as a photobiont. *Caloplaca* is a cosmopolitan lichen genus found in most xeric and mesic habitats [57].

The clones retrieved from lichen samples are shown in Table 2. The three lichens showed *Trebouxia aggregata* as the photobiont, which confirmed that the finding of this green alga in the Cross fragments was associated to the occurrence of a lichenic crust. Muggia et al. [56] stated that the photobiont of *Caloplaca variabilis* and other closely related *Caloplaca* spp. (e.g., *Caloplaca chalybaea*) was an unknown taxon of *Trebouxia*, typical of the Mediterranean area. These three lichens (among others) identified in the City Hall were also found on natural limestone outcrops, buildings and monuments [58–62].

**Table 2.** Identification of eukaryotic organisms associated with the three main lichens colonizing the limestones from Seville City Council.

| Lichen | Associated Species | Accession Number | Group * |
|---|---|---|---|
| *Kuettlingeria teicholyta* | *Trebouxia aggregata* | ON479841 | *Chlorophyta* |
| | *Pyrenodesmia chalybaea* | ON479847 | *Lichen* |
| | *Xanthoria parietina* | ON479846 | *Lichen* |
| | *Xanthoria* sp. | ON479845 | *Lichen* |
| | *Phoma* sp. | ON479844 | *Fungi* |
| | *Ceratobasidium* sp. | ON479843 | *Fungi* |
| | *Pyrenidium* cf. *actinellum* | ON479842 | *Fungi* |
| | *Chaetothyriales* sp. | ON479848 | *Fungi* |
| | *Rhinocladiella* sp. | ON479849 | *Fungi* |
| *Caloplaca citrina* group | *Trebouxia aggregata* | ON479852 | *Chlorophyta* |
| | *Xanthoria parietina* | ON479850 | *Lichen* |
| | *Xanthoria* sp. | ON479851 | *Lichen* |
| *Pyrenodesmia variabilis* | *Trebouxia aggregata* | ON479854 | *Chlorophyta* |
| | *Capnobotriella* sp. | ON479853 | *Fungi* |

* Lichens are taxonomically classified by their fungal partner, so all lichens belong to the *Fungi* kingdom. However, in this work we distinguish between fungi and lichens for convenience.

Clones of other lichens (*Pyrenodesmia chalybaea*, *Xanthoria parietina* and *Xanthoria* sp.) were present in the sample of *Kuettlingeria teicholyta*, and in the *Caloplaca citrina* group lichen were identified as being *Xanthoria parietina* and *Xanthoria* sp.

In addition, a few fungi were retrieved from *Kuettlingeria teicholyta* (*Pyrenidium* cf. *actinellum*, *Phoma* sp., *Rhinocladiella* sp., *Ceratobasidium* sp., and *Chaetothyriales* sp.). Most of these fungi are lichenicolous from arid habitats (e.g., *Phoma*, *Pyrenidium*, *Rhinocladiella*), as well as *Capnobotriella* sp., associated with *Pyrenodesmia variabilis* [49–51,63,64].

Table 2 shows that a complex crustose lichen community colonizes the building façade of the Seville City Hall. This community is the result of the biological regeneration after the restoration of the Cross and façade, carried out between 2008 and 2010. The protocol consisted in mechanical cleaning of the limestone surfaces with water and natural soap and the joints were filled with lime mortar [65].

Previous studies on the efficacy of mechanical procedures and laser cleaning for removal of crustose lichens from granite showed that these methods were unable to eliminate the thalli into fissures [66,67]. In our case, Figures 5 and 6 showed that different organisms penetrate the limestone through pitting and channels. Likely, the 2008–2010 cleaning did not affect the endolithic hyphae, which grow from the crustose lichen thalli and penetrate the porous limestone. These endolithic hyphae facilitate the subsequent recovery of the lichen community, in the same way that the roots in the soil allow the regeneration of the aerial part of the plants after harvesting.

Favero-Longo and Viles [68] recognized that rock-dwelling organisms are agents of the deterioration of cultural heritage stone surfaces, but their removal is usually followed by a rapid recolonization. Ariño et al. [59] reported that epilithic lichens can provide a protective cover for stones because biodeterioration is a slower process than physical and chemical weathering produced by environmental factors. This fact was confirmed in a study on colonized and uncolonized flagstones in an archaeological site. Concha-Lozano et al. [69] found that limestone secondary porosity was filled by lichen hyphae, and the thalli gave a waterproofing effect that conferred preservation to ancient monuments. Pinna [70] stated that a lichen crust protects porous stones from weathering by stabilizing the surfaces, and although the removal of lichens from statues and monuments is widely practiced, it can damage the stone due to the intimate association of biological structures with the stone components.

Salvadori and Municchia [71] in a review on the role of fungi and lichens on monuments stated that biodeterioration and bioprotection are not mutually exclusive and can occur simultaneously in a lichen community.

Indeed, Nascimbene et al. [72] studied the re-colonization of a limestone statue by lichens 12 years after cleaning and restoration. The lichen flora was dominated by nitrophilic species of *Caloplaca* and *Verrucaria*, which gave an orange–grey–black color to the community. The restoration protocol of the statue included the application of a biocide (Metatin N-58-10/101), mechanical removal of the biomass, a new application onto the cleaned surfaces and consolidation with Akeogard CO. The authors reported that the long-term effectiveness of the restoration was low because the total number of species (25) before restoration was similar to that found years after (20), indicating a progressive colonization along the time. This colonization was favored by the location of the statue surrounded by trees in a park [73]. In the case of the restored Cross of the Inquisition, located in an urban environment with scarce trees in the surroundings, it was expected that recolonization by ascospores from epiphytic lichens will take place in the very long-term.

A few reports suggested that the application of biocides, water-repellent products and consolidants was an effective treatment for preventing biological growth on stones [74–78], as performed in this restoration. However, due to the short distance of the lichen community colonizing the façade of the City Hall (Figure 7) to the Cross of Inquisition, the dispersal and settlement of ascospores on the restored limestone cannot be ruled out.

### 3.4. Restoration of the Cross of the Inquisition

The restoration of the Cross was carried out by Atelier Samthiago, Conservaçao e Restauro. The result of the final restoration is shown in Figure 1C. The main steps included:

*Photographic survey*: mapping of pathologies, registration of existing pathologies, three-dimensional record.

*Dry cleaning of superficial dirtiness*: removal of superficial dirt deposited on the monument's surfaces, using soft brushes and spatulas.

*Biocide treatment*: Preventol RI80 was applied with a soft paintbrush and by spraying on the entire area including the pedestal.

*Physico-chemical cleaning of surfaces*: after biocide treatment, a cleaning with nylon brushes and water was applied. For the homogenization of the surface layer with chromatic alteration, a poultice soaked with an aqueous solution of ammonium bicarbonate, EDTA and drops of neutral Teepol detergent was applied. In the cruise and the pedestal, poultices with different products and aqueous solutions were applied.

*Consolidation of the stone surface*: to reinforce the inter-granular cohesion of the stone, which was very weakened in some areas, all the fragments and the pedestal of the cross were subjected to a general consolidation with Estel 1000, applied with a brush. The consolidation was reinforced in areas of cracks, applying the product with a syringe.

*Elimination and replacement of corroded internal metallic structures*: removal of resin remains from previous restorations and metal elements that lost their structural function.

*Volumetric reconstitution*: adhesion of the fragments allowing volumetric reconstitution and reintegration of structural gaps with Sikadur 31EF epoxy resin adhesive. In some joints, the adhesion was reinforced with small fiberglass or stainless steel spigots.

*Coating and micro-coating for the reintegration of surface discontinuities*: application of lime putty with chromatic correction for coating and micro-coating of surface gaps, fissures and joints. Lime putty was used in a 1:1 ratio between inert hydraulic lime binder and lime Glue Bio Pasta Gordillos, adding natural pigments.

*Opening, cleaning and sealing of joints, with volumetric correction*: the cement filling the joints was mechanically removed using a chisel, then cleaned with nylon brushes and water in a controlled mode. The joints were cleaned and reintegrated with the same lime putty.

*Assembling the cross on the cruise*: fixation of the cross on the cruise with epoxy resin (Sikadur 31EF) by means of a metallic spigot treated and reinforced with two more fiberglass spikes of 8 mm diameter and about 90 mm long, distributed between the two elements.

*Final protection*: the water-repellent and anti-graffiti product Aguasil ST was applied with a brush to the entire monument surface, paying special attention to the carving, the cruise and the pedestal.

## 4. Conclusions

The data obtained are consistent with the exposure of the Cross of the Inquisition to an urban environment, where it has remained for more than 100 years. During that time, a lichen community colonized the limestone, associated with bacteria, fungi and bryophytes. Evidences of ancient (pitting) and recent (fungal hyphae) colonizations have been found. The occurrence of lichens, as denoted also by the green algae photobiont, lichenicolous and black fungi, and other microorganisms in the limestone have composed a mature and well-developed urban community on the limestone over the years, even after the 2008–2010 restoration, due to its regeneration capacity. Likewise, the species found in the Cross limestone were also identified in other urban environments and monuments, located in the Mediterranean Basin.

In arid and semi-arid environments of the Mediterranean Basin the climatic conditions are too extreme for most fungi, so the communities move towards the so-called black yeasts and other rock-inhabiting fungi, which form small black colonies on the surface and within the limestone and often occur in close association with lichens.

The microbial activity has been detrimental for the integrity of the limestone. The activity of rhizines (hyphae that keep the lichen fixed to the substratum), the rhizoids of bryophytes, and endolithic microorganisms has produced channels and pitting, which have evidenced the lytic activity of microorganisms on the stone surface. Therefore, stone consolidation was achieved with Estel 1000. In addition, a biocide (Preventol RI80), able to penetrate the porous limestone and active against bacteria, fungi, lichens and bryophytes, was used.

**Author Contributions:** Conceptualization, C.S.-J.; methodology, C.S.-J. and C.C.; investigation, V.J.; J.L.G.-P., A.G.-B., J.C.C. and S.S.-M.; restoration, C.C.; writing—original draft preparation, C.S.-J. and J.C.C.; writing—review and editing, C.S.-J. All authors have read and agreed to the published version of the manuscript.

**Funding:** The research was funded by Atelier Samthiago and the restoration of the Cross of the Inquisition was supported by the Seville City Hall.

**Institutional Review Board Statement:** Not applicable.

**Informed Consent Statement:** Not applicable.

**Data Availability Statement:** The nucleotide sequences were deposited into the NCBI GenBank database under accession numbers ON479828–ON479853.

**Acknowledgments:** The authors wish to acknowledge the professional support of the CSIC Interdisciplinary Thematic Platform Open Heritage: Research and Society (PTI-PAIS).

**Conflicts of Interest:** The authors declare no conflict of interest.

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
