# Peer review of "Holistic Approach to the Restoration of a Vandalized Monument: The Cross of the Inquisition, Seville City Hall, Spain"

_applsci, doi:10.3390/app12126222_

Round 1

Reviewer 1 Report

Review of the manuscript

Title: Holistic Approach to the Restoration of a Vandalized Monu-2 ment. The Cross of the Inquisition, Seville City Hall, Spain

The work describes in a clear and well-structured way all the phases and the various topics covered.

An exhaustive bibliography frames the issues addressed and accompanies the drafting. The analyses carried out and the results are clear and well described. The work is very interesting.

For the reviewer, the work is accepted.

Author Response

Thank you for the comments

Reviewer 2 Report

The electron microscopy methodology should be better explained and the captions of the corresponding figures should be improved. 

Author Response

Electron microscopy methodology and figure legends improved

Reviewer 3 Report

I think that the paper is well written and it covers the topic that will be interesting to all restaurateurs facing the problem of vandalism and restauration of the monument. The used restoration procedures and their effects are clearly presented, so that in can be used for all similar restoration processes.

Author Response

Thank you for the comments